# Inhibition of β-Catenin Activity Abolishes LKB1 Loss-Driven Pancreatic Cystadenoma in Mice

**DOI:** 10.3390/ijms22094649

**Published:** 2021-04-28

**Authors:** Mei-Jen Hsieh, Ching-Chieh Weng, Yu-Chun Lin, Chia-Chen Wu, Li-Tzong Chen, Kuang-Hung Cheng

**Affiliations:** 1Institute of Biomedical Sciences, National Sun Yat-Sen University, Kaohsiung 804, Taiwan; rohsiehs@yahoo.com.tw (M.-J.H.); inoursky@gmail.com (C.-C.W.); yoyolin322@gmail.com (Y.-C.L.); so62485@gmail.com (C.-C.W.); 2Division of Neurology, Department of Internal Medicine, Kaohsiung Armed Forces General Hospital, Kaohsiung 802, Taiwan; 3National Institute of Cancer Research, National Health Research Institutes, Tainan 704, Taiwan; 4Department of Internal Medicine, Kaohsiung Medical University Hospital, Kaohsiung Medical University, Kaohsiung 807, Taiwan; 5Department of Oncology, National Cheng Kung University Hospital, National Cheng Kung University, Tainan 704, Taiwan; 6Regenerative Medicine and Cell Therapy Research Center, Kaohsiung Medical University, Kaohsiung 807, Taiwan; 7Department of Medical Laboratory Science and Biotechnology, Kaohsiung Medical University, Kaohsiung 807, Taiwan

**Keywords:** pancreatic cancer, LKB1, Wnt/β-catenin, MCN

## Abstract

Pancreatic cancer (PC) is the seventh leading cause of cancer death worldwide, and remains one of our most recalcitrant and dismal diseases. In contrast to many other malignancies, there has not been a significant improvement in patient survival over the past decade. Despite advances in our understanding of the genetic alterations associated with this disease, an incomplete understanding of the underlying biology and lack of suitable animal models have hampered efforts to develop more effective therapies. LKB1 is a tumor suppressor that functions as a primary upstream kinase of adenine monophosphate-activated protein kinase (AMPK), which is an important mediator in the regulation of cell growth and epithelial polarity pathways. LKB1 is mutated in a significant number of Peutz–Jeghers syndrome (PJS) patients and in a small proportion of sporadic cancers, including PC; however, little is known about how LKB1 loss contributes to PC development. Here, we report that a reduction in Wnt/β-catenin activity is associated with LKB1 tumor-suppressive properties in PC. Remarkably, in vivo functional analyses of β-catenin in the Pdx-1-Cre LKB1^L/L^ β-catenin^L/L^ mouse model compared to LKB1 loss-driven cystadenoma demonstrate that the loss of β-catenin impairs cystadenoma development in the pancreas of Pdx-1Cre LKB1^L/L^ mice and dramatically restores the normal development and functions of the pancreas. This study further determined the in vivo and in vitro therapeutic efficacy of the β-catenin inhibitor FH535 in suppressing LKB1 loss-driven cystadenoma and reducing PC progression that delineates the potential roles of Wnt/β-catenin signaling in PC harboring LKB1 deficiency.

## 1. Introduction

Pancreatic cancer (PC) is the seventh leading cause of cancer death in both sexes worldwide [1,2]. With a 5-year survival rate of only 8% and a median survival of less than 6 months, a diagnosis of pancreatic adenocarcinoma carries one of the most dismal prognoses in the medical field. Due to a lack of specific symptoms and limitations in diagnostic methods, the disease often eludes detection during its formative stages. Unlike other malignancies, no marked improvement has been achieved in PC survival [3]. The signature molecular alterations in PC include multiple evolutionary steps of the precursor lesions of which progression involves the acquisition of mutations in Kras, Ink4a, p53, SMAD4, LKB1, APC, or β-catenin [4,5,6,7,8]. Pathology analysis has revealed that PC appears to arise from three distinct precursor lesions that transform the pancreatic ducts: pancreatic intraepithelial neoplasms (PanINs), which are small and focal; intraductal papillary mucinous neoplasms (IPMNs), which are moderate-sized, papillary cystic lesions lined by mucin-producing tall columnar epithelium; and mucinous cystic neoplasms (MCNs), comprising oligomegacysts with a single thin layer of cuboidal and flattened epithelium and associated with an ovarian-type of stroma [9,10,11]. These lesions present distinct histological morphologies and clinical significance but share a common mutation profile. The basis of these structural and histological differences is unknown, but may be associated with the cell of origin, various mutation combinations, the order of the mutational events, or other factors [6].

Liver kinase B1 (LKB1), also known as STK11, is a serine–threonine kinase whose mutation is responsible for Peutz–Jeghers Syndrome, which is a hereditary condition that results in the development of benign (hamartomatous) polyps in the gastrointestinal tract, mucocutaneous pigmentation, and a predisposition to developing cancers in a variety of tissues: the colon, small intestine, breast, ovary, pancreas, and lung [12,13,14,15,16]. The LKB1 protein forms a heterotrimeric complex via interacting with two proteins—STE20-like pseudokinase, named the Ste20-related adaptor (STRAD) and a third protein named mouse protein 25 (MO25)—to mediate downstream intracellular signaling cascades [17]. LKB1 works as a primary upstream kinase of adenine monophosphate-activated protein kinase (AMPK), which is a necessary element in cell metabolism that is required for mediating cell proliferation and energy homeostasis [18,19]. Additionally, the LKB1-AMPK pathway has been implicated in the regulation of multiple pathways and participates in many biological stress responses, including chromatin remodeling, angiogenesis, p53-dependent apoptosis, cell cycle arrest, the DNA damage response, cellular polarity, and differentiation [20,21,22,23]. Apart from the above-described aspects, LKB1 has also been suggested to possess a variety of potentially tumor-suppressive functions, such as the inhibition of mammalian targets of rapamycin, regulation of epithelial–mesenchymal transition (EMT), and inhibition of cell cycle progression [24,25]. Recently, many investigations have demonstrated that the somatic deletion of LKB1 linked to lung tumorigenesis and its inactivation is a common event in adenocarcinomas (34%) and squamous cell carcinomas (19%) of the lung to further unleash its tumor-suppressive activity [26,27]. In addition, a somatic mutation of LKB1 has been reported in pancreatic, liver, and biliary cancers, as well as malignant melanomas [28,29,30,31,32]. Although many studies have reported the potential roles of the LKB1 tumor-suppressor gene in human cancers, including non-small cell lung cancer (NSCLC), little is known about how LKB1 signaling regulates the pancreatic carcinogenesis pathways and reprograms the PC microenvironment. To understand LKB1-mediated tumor suppression in pancreatic tumorigenesis, we first utilized the pancreatic progenitor-specific Pdx-1 gene promoter-driven Cre recombinase strain to cross with conditional LKB1 loxp/loxp knockout mice, in order to investigate the developmental functions of LKB1 in pancreatic organogenesis and to examine whether pancreas-specific LKB1 deletion triggers pancreatic carcinogenesis in vivo [33,34]. In this study, we dissected the potential key players involved in the development of pancreatic cystic lesions in pancreatic conditional LKB1 null mice.

## 2. Results

### 2.1. LKB1 Loss Specific to the Pancreas Leads to Severe Defects in Epithelial Cell Polarity and the Gradual Development of Mucin Cystadenoma of the Pancreas in Mice

Recently, we reported the pancreatic phenotype of the aberrant expression and deficiencies in APC and/or p53. This work served to address issues of genetic alterations and the tumor suppression function in the development of pancreatic cysteadenoma [35]. Additionally, previous studies reported that LKB1 depletion in the pancreas provokes metaplastic changes of acinar cells into ductal structures (tubular metaplasia) and also forms mucin cystadenoma. Pdx-1Cre LKB1^L/L^ mutant mice possessing these compound mutant alleles experienced transformation of the ductal complexes into classical benign pancreatic mucin cystadenoma (MCN) [36]. In the present study, we first crossed the conditional LKB1 floxed mouse with a pancreatic-specific genetically engineered transgenic line. The Pdx-1Cre strain exerts the pancreas-specific knockout of LKB1 during pancreatic organogenesis at approximately embryonic day (E) 8.5–9.5 (Figure 1A). The Pdx-1Cre LKB1^L/L^ mouse model is predisposed to MCNs of the pancreas, with a nearly complete penetrance. Later, we further immunohistologically characterized cyst adenoma and other early lesions derived from the Pdx1-Cre LKB1^L/L^ strain using an immunohistochemistry (IHC)-based method in order to obtain clues for profiling the lesion’s characters and to identify essential signaling pathways affected by LKB1 depletion in vivo. Kaplan–Meier survival curves for Pdx-1Cre LKB1^L/L^ mice and their wild-type controls revealed that Pdx-1Cre LKB1^L/L^ mice (*n* > 16) displayed a significantly shorter lifespan compared to wild-type ones (Figure 1B). We also observed that Pdx-1Cre LKB1^L/L^ mutant mice exhibited a reduced body weight after 5 weeks of age when compared to wild-type mice (data not shown). Additionally, we detected abnormal glucose tolerance tests (GTT) in Pdx-1Cre LKB1^L/L^ (PL) mice compared to wild-type mice (Figure 1C). As shown in Figure 1D, LKB1 deficiency mutant mice developed early ductal lesions reminiscent of human MCN, and tumor histological features were shown by hematoxylin and eosin (H&E) staining (Figure 1E).

Next, we performed a series of immunohistochemistry (IHC) staining analyses on these cystic lesions using markers of differentiated lineages in the pancreas, as well as for some developmental and cancer-associated markers. We showed that the mouse MCNs stained positively for markers of ductal epithelia—cytokeratin 19 (CK19)—consistent with the possibility that they may derive from ductal epithelial cells directly or from their progenitors. Of note, they were negative for the acinar cell marker elastase and the islet β cell marker insulin (Figure 1F). One recent work by Makoto and his colleagues suggested that the activation of Wnt signaling within the stroma might contribute to the development of human pancreatic mucinous cystic neoplasms [37]. Therefore, we explored the link between loss of LKB1 and activation of Wnt signaling in our Pdx-Cre-driven LKB1 loss mouse model of MCN. Here, we identified that the mouse MCN lesions stained positively for Wnt signaling pathways. These cystic lesions exhibited an increased expression of Ki67 and total and active β-catenin activities compared to the surrounding normal pancreas, as shown by IHC analysis (Figure 1Gi–ii). Of note, our findings confirmed that that the active β-catenin (ABC) positive nuclear staining can occur within the cystic epithelial tumors and stroma cells, which under the light of the Wnt signaling pathway potential crosstalk between tumor and stroma in MCN lesions.

### 2.2. LKB1 Ablation Stimulates the Activation of β-Catenin Signaling in the Mouse Pancreas and Results in the Development of Pancreatic Cystadenoma

In the pancreas of wild-type mice, β-catenin was predominantly observed at the membrane of islets and acinar cells. Moderate or strong nuclear β-catenin expression levels were found during PanINs progression and pancreatic ductal adenocarcinoma (PDAC) formation; however, a significant increase in the β-catenin nuclear staining intensity was detected in the PDAC of Pdx-1Cre LSL-Kras^G12D^ p53^L/+^ mice (Figure 2A). In this LKB1 loss-driven MCN model, we also observed the inactivation of LKB1 accompanying an increasing active β-catenin protein level (Figure 2A). One hypothesis offered to explain this phenomenon is that the transcriptional regulation causes an increased expression of β-catenin mRNA in LKB1-deficient PDAC cells; another could be that LKB1 kinase is involved in the modification of Wnt/β-catenin activity and regulates the stability of the β-catenin protein. However, our real-time PCR results suggested that the depletion of LKB1 does not increase the mRNA level of β-catenin in PDAC cells (Figure 2B). Our data further supported the idea that LKB1 deficiency leads to a stabilized/active β-catenin protein to prevent β-catenin degradation, which results in decreasing β-catenin degradation and maintains the activation of Wnt signaling in PDAC cells (Figure 2C). Intriguingly, our previous study demonstrated that LKB1 regulates GSK3β phosphorylation through the LKB1–APC–GSK3β interaction and influences the activity of the Wnt signaling pathway in lung cancer [38]. Additional work is required to further elucidate the detailed mechanisms underlying the role of LKB1 in canonical and/or non-canonical Wnt signaling pathways, either during the onset of pancreatic cystadenoma or progression to metastatic malignancy.

### 2.3. Conditional Inactivation of the β-Catenin Gene Does Not Affect Normal Pancreas Development and Function

To characterize the functional roles of β-catenin deficiency in the development of the pancreas, we obtained β-catenin flox/flox mice developed in the laboratory of Dr. Ting-Fen Tsai at Yang Ming University, Taiwan, in which the conditional β-catenin loxp/loxp allele was engineered to sustain the Cre-mediated excision of exon 2 to exon 6 (Figure 3A) [39]. We then crossed β-catenin flox/flox (β-catenin^L/L^) mice with Pdx-1Cre transgenic mice to direct Cre recombinase expression specific in the endodermal progenitor lineages of the pancreas (Pdx-1-positive subpopulation). The Cre recombinase-induced DNA rearrangement of the β-catenin gene locus in the pancreas from Pdx-1Cre β-catenin^L/L^ mutant mice was determined by allele-specific PCR genotyping and by the detection of β-catenin protein expression in Western blot and/or IHC analysis (Figure 3B) to confirm the specific delete β-catenin expression in the pancreas. In this experimental setting, we reported that Pdx-1Cre β-catenin^L/L^ mice (*n* = 14) were born at the expected Mendelian ratio, indicating that the Pdx-1Cre-mediated inactivation of β-catenin in the pancreas did not cause embryonic lethality. Kaplan–Meier survival curves for Pdx-1Cre β-catenin^L/L^ mice and their wild-type Pdx-1Cre controls revealed that Pdx-1Cre β-catenin^L/L^ mice displayed a similar lifespan to wild-type ones (Figure 3C). A functional analysis of glucose tolerance (GTT) also revealed no significant differences in the Pdx-1Cre β-catenin^L/L^ mice in comparison to wild-type controls (Figure 3D). The appearance of the pancreas, body weight, and pancreas weight did not reveal any abnormality in Pdx-1Cre β-catenin^L/L^ mutant mice compared to wild-type mice (Figure 3E and data not shown). Histological analysis of pancreatic tissues of Pdx-1Cre β-catenin^L/L^ mice at early and late time points showed normal exocrine granular and ductal structural components and islet formation after verification that nuclear β-catenin expression was completely lost in the pancreas of Pdx-1Cre β-catenin^L/L^ mice (Figure 3F,G). Meanwhile, IHC analysis confirmed the loss of β-catenin protein expression in the pancreas of Pdx-1Cre β-catenin^L/L^ mice and demonstrated that the staining Ki67 patterns were very similar between Pdx-1Cre β-catenin^L/L^ mutant and Pdx-1Cre control mice (Figure 3Gi–ii). These data revealed that β-catenin inactivation does not prominently affect pancreas development or play any role in the induction of pancreatic carcinogenesis.

### 2.4. Conditional Loss of β-Catenin Suppresses LKB1-Deficient-Mediated Pancreatic Tumorigenesis in Mice

As noted above, Wnt signaling is thought to have highly context-dependent effects on this tumorigenesis. In initial investigations of this pathway in MCN pathogenesis, we further confirmed that the aberrantly increased nuclear β-catenin activity existing in the cystic tumors of the Pdx-1CreLKB1^L/L^ mice (Figure 2A). To determine whether the depletion of β-catenin activity can abolish MCN formation in Pdx-1CreLKB1^L/L^ mice, we further produced triple mutant mice (Pdx-1Cre LKB1^L/L^ β-catenin^L/L^ mice) by crossing the mice with a conditional β-catenin allele with Pdx1-Cre LKB1^L/L^ animals which had a well-defined course of cystadenoma initiation and progression. Surprisingly, we observed that Pdx-1CreLKB1^L/L^ β-catenin^L/L^ mice displayed a normal pancreas architecture and function in our aged cohort of Pdx-1CreLKB1^L/L^ β-catenin^L/L^ mice. Pdx-1CreLKB1^L/L^ β-catenin^L/L^ mice displayed no evidence of any gross anatomic or physiological abnormalities of the pancreas and exhibited normal pancreatic cytoarchitecture and normal differentiation throughout our aged cohort study (Figure 4A). Unlike Pdx-1CreLKB1^L/L^ mice, the Kaplan–Meier survival curves of Pdx-1Cre LKB1^L/L^ β-catenin^L/L^ and Pdx-1Cre wild-type mice revealed the same level of life expectancy. (Figure 4B). Figure 4C shows that Pdx-1CreLKB1^L/L^ β-catenin^L/L^ mice restored the normal fasting blood glucose level within 2 h, which remained significantly high in Pdx-1CreLKB1^L/L^ mice. Unlike the IHC staining patterns in cystic neoplasms of the pancreas of Pdx-1Cre LKB1^L/L^ mice, the IHC analysis for amylase, insulin, and glucagon showed normal expression, and staining with CK19 confirmed the presence of a normal proportion of pancreatic ducts in Pdx-1Cre LKB1^L/L^ β-catenin^L/L^ mice (Figure 4D). Additionally, the relative size and distribution of islets, as well as the architecture of the exocrine acinar tissue of Pdx-1CreLKB1^L/L^ β-catenin^L/L^ mice, were indistinguishable from those observed in the pancreas from wild-type littermates (Figure 4D,Ei–ii).

### 2.5. The Wnt Inhibitor FH535 Inhibits MCN Formation in Pdx-1CreLKB1^L/L^ Mice

To further confirm whether LKB1 loss-induced MCN development is Wnt/β-catenin signaling pathway-dependent, the Wnt inhibitor FH535 was then selected to treat the MCN model of Pdx-1Cre LKB1^L/L^ compound mice, as FH535 can inhibit Wnt/β-catenin signaling activation. We conducted in vivo testing to examine whether blocking the activation of Wnt/β-catenin signaling with FH535 is sufficient for eradicating the LKB1 loss-caused formation of abnormal cystic pancreatic lesions in mice. Therefore, we treated Pdx-1Cre LKB1^L/L^ mice with FH535 (15 mg/kg) over a time period of 6 weeks, starting at an age of 6 weeks. For controls, mice were treated with DMSO (vehicle). At the end of treatment, 12-week-old mice from each group were sacrificed and the pancreata were harvested for macro and pathohistological examination. We observed a significant extension of MCN lesion-free survival in all mice treated with FH535 and there was no sign of any MCN lesions in FH535-treated Pdx-1Cre LKB1^L/L^ mice (Figure 5A). Of note, the functional analysis for glucose tolerance tests (GTT) revealed that FH535 can exert a hypoglycemic effect, and appears to significantly improve the fasting blood glucose level in Pdx-1Cre LKB1^L/L^ mice (Figure 5B). Furthermore, Western blot analysis confirmed significant reductions in the downstream Wnt signaling pathway effector proteins, such as β-catenin and cyclin D1, in response to FH535 treatment in vivo (Figure 5C). Our data demonstrated decreased levels of total and active β−catenin (ABC) and cyclin D in Pdx-1CreLKB1^L/L^ mice treated with FH535 compared to those of DMSO controls (Figure 5C). At the endpoint (week 12), histological analysis of pancreatic tissues of Pdx-1Cre LKB1^L/L^ mice after FH535 treatment exhibited normal exocrine granular, islet, and ductal structural components in the pancreas. IHC analysis revealed that amylase, insulin, glucagon, and pancreatic ductal marker CK19 exhibited normal re-expression in FH535-treated Pdx-1Cre LKB1^L/L^ mice (Figure 5D). In contrast, amylase, insulin, glucagon, and CK19 expression was largely absent in neoplasm in the pancreas of Pdx-1CreLKB1^L/L^ mice that received the vehicle (DMSO) (Figure 5D). Subsequently, IHC analysis demonstrated that FH535 treatment resulted in reduced levels of Ki67, total β-catenin, and ABC expression in pancreata of the Pdx-1CreLKB1^L/L^ model (Figure 5E). Lastly, mouse cytokine array analysis also confirmed that pancreata lysates from Pdx-1CreLKB1^L/L^ mice treated with FH535 reduced several inflammation-related chemokines and cytokine production, such as CCL1, CCL2, CCL27, CXCL16, LIX, TNF RI RII and IL-4, compared to the Pdx-1CreLKB1^L/L^ (DMSO) group (Figure 5F). Thus, there are clearly many cytokines and chemokines that are upregulated by LKB1 loss in the pancreas, and are relevant to the pathogenesis of pancreatic MCNs. Most need further investigation to confirm their specific impact on MCN pathology and clinical features, and whether they may be therapeutically targeted using in vivo models.

### 2.6. shRNA Knockdown of LKB1 Expression Increases Cell Proliferation and the Migration of PDAC Cells In Vitro

Next, we evaluated whether LKB1 knockdown affects human PDAC tumorigenic properties. To do so, we first screened several human PDAC cell lines, including 6 PDAC cell lines (Panc-1, AsPC-1, BxPC3, CFPAC, Hs766T, and MiaPaCa2), and immortalized normal human pancreatic ductal cell line (HPDEC) served as a positive control for LKB1 expression to detect their LKB1 expression statuses. As shown in Figure 6A, our data demonstrated that different expression levels of LKB1 in a variety of human PDAC cells, whereas high LKB1 protein expression was detected in Panc-1, AsPC-1, and HPDEC control cells by Western blot analysis. Next, we used shRNA to stabilize knockdown LKB1 expression in Panc-1 and AsPC-1 human PDAC cell lines. Panc-1 and AsPC-1 cells were transfected with the LKB1 shRNA lentiviral vector or mock control vector (sheGFP), followed by selection with puromycin for two weeks, in order to obtain stable shLKB1 knockdown stable clones. Our Western blot analysis confirmed that shLKB1 effectively suppressed LKB1 expression in Panc-1 and AsPC-1 cells, whereas the control pLKO1 eGFP cells displayed no effect (Figure 6B). We then detected whether the knockdown of LKB1 affects the proliferation of human PDAC cells in vitro. We found that shLKB1 knockdown substantially increased the proliferation of Panc-1 and AsPC-1 cells by 1.5 fold compared to the controls by using cell proliferation assays (*p* < 0.001; Figure 6C). In addition, the knockdown of LKB1 in Panc-1 or AsPC-1 PDAC cells displayed a greater ability for colony formation in vitro (Figure 6D). Therefore, the specific inhibition of LKB1 in human PDAC cells significantly increased cell proliferation rates and tumorigenicity in vitro.

Meanwhile, to examine the effect of LKB1 knockdown on the PDAC cell migratory ability, we performed an in vitro wound healing assay to compare the migratory capacities of LKB1 shRNA and control Panc-1 and AsPC-1 human PDAC cells. After overnight incubation, our results indicated that the knockdown of LKB1 (by LKB1 shRNA) in Panc-1 and AsPC-1 cells increased PDAC cell migration (Figure 6E). In summary, our results indicated that LKB1 may be involved in mediating cell proliferation and migration in PDAC cells. Most importantly, we demonstrated that the knockdown of LKB1 in Panc-1 cells significantly induces the activation of Wnt signaling pathway by using Western blotting or a TOP-FOP luciferase reporter assay (Figure 6F,G). Later, in the in vitro study of the FH535 inhibitor, we used a small molecule inhibitor of canonical Wnt signaling, called FH535, to further demonstrate that FH535 reduces cell proliferation in Panc-1 and AsPC-1 shLKB1 cells (Figure 6H). Therefore, the inactivation of LKB1 indeed enhances canonical Wnt signaling in PDAC and leads to increased cell proliferation and migration of PDAC.

## 3. Discussion

PC is difficult to treat, with conventional chemotherapy of gemcitabine only affording a few months of incremental survival [40]. Reflecting on the history of failed clinical trials including chemotherapy, improvements in treating PC will emerge from a better knowledge of the mechanisms of not only the derangements in and consequently acquired capabilities of the tumor cells, but also in the aberrant microenvironment that becomes established to suppress the host anti-tumor immune response and to support, sustain, and enhance neoplastic progression [41]. By combining rigorous mechanistic investigations of the tumor microenvironment and its genetic controls with experimental therapeutic trials that leverage both resultant knowledge of mechanisms and the de novo development and progression of pancreatic cancer in genetically engineered mouse models, we anticipate that new therapeutic strategies will be suggested, motivating new clinical trials against PC.

LKB1, also named serine/threonine-protein kinase (STK11), functions as a major regulator of energy homeostasis through the activation of AMP-activated protein kinase (AMPK) and plays an important role in vascular development, cell polarity, cell growth, and tumor suppression [19]. In humans, germline mutations of LKB1 are associated with Peutz–Jeghers syndrome (PJS), benign gastrointestinal polyps (hamartomas), and a ∼30-fold increased risk of diagnosis with gastrointestinal malignancy at age >60 [42,43,44]. In PJS, there are a range of pancreatic neoplasms, including pancreatic ductal adenocarcinoma and two types of cystic tumors, intraductal papillary mucinous neoplasia and serous cystadenoma, and mixed types have now been reported [36]. LKB1 is one of the most commonly mutated genes across a range of sporadic tumors, including non-small cell lung, cervical, skin, and pancreatic carcinoma [45]. Somatic mutations of LKB1 and LKB1 loss of function have been identified and initially reported in certain sporadic human cancers, with an especially high frequency in lung cancer [30,46]. LKB1 can regulate the activity of AMPK to control critical cell metabolism and growth. AMPK phosphorylates and activates TSC2, which is a negative regulator, to decrease protein synthesis by inhibiting the mTOR pathway. The LKB1-AMPK pathway also inhibits cancer metastasis by suppressing epithelial–mesenchymal transition (EMT). In addition, although the AMPK pathway in tumorigenesis has conflicting roles in cancers, several pieces of accumulated evidence have indicated that AMPK can function as a pro-tumorigenic regulator by promoting autophagy and increasing glucose uptake [47]. Therefore, AMPK activation has been specifically associated with the progression of several different cancer types, including breast cancer, prostate cancer, colon cancer, and hepatocellular cancer [48]. In a recent study, LKB1 was also found to be somatically mutated in cervical carcinoma, which is caused by the human papillomavirus [49]. Meanwhile, several lines of evidence have shown that a low expression of LKB1 is associated with significantly worse overall survival compared with patients with LKB1 high-expression tumors in PDAC [50]. Although the inactivation of LKB1 has been reported in only 4–7% of sporadic PDAC, a low LKB1 expression level occurs in 20–25% of PDAC and negatively impacts the clinical outcomes of PDAC patients [50]. Importantly, the utility of LKB1 gene targeting in mouse genetic studies has demonstrated that the targeted disruption of both LKB1 alleles leads to embryonic lethality at the middle gestation stage. Conventional knockout *LKB1* (+/−) mice develop gastrointestinal polyps, of which the histological characteristics resemble those of the Peutz–Jeghers syndrome hamartomas, and more than 70% of male mice develop hepatocellular carcinomas (HCCs) in 50 weeks [51]. Meanwhile, Peutz–Jeghers syndrome is characterized by hamartomatous gastrointestinal polyposis, and LKB1 mutation combined with mutations of the β-catenin gene and p53 gene can convert hamartomatous polyps into early gastric adenoma and carcinoma [52]. Concurrently, Collet and his colleagues reported that Kras mutations synergize with LKB1 inactivation, and lead to the development of IPMN in mice. However, based on their findings, they concluded that the lack of β-catenin did not impede the formation of intraductal papillae and their progression to papillary lesions in IPMN, which probably because the activating Kras^G12D^ mutation combined with LKB1 ablation produced more synergistic effects in promoting development of IPMN [53].

In this study, we first immunohistochemically characterized the specific LKB1 deletion in the pancreas of the mouse and demonstrated that this results in the development of cystadenocarcinoma, and we reported, for the first time, that the activation of the Wnt signaling pathway and increased nuclear β-catenin activity are the crucial signaling events required for the growth and maintenance of murine MCN in Pdx-1CreLKB1^L/L^ mice. This is consistent with our previous report, in which we demonstrated that loss of APC function results in the induction of pancreatic MCN formation in the context of p53 loss [35]. Of note, Makoto and his colleagues showed that activated WNT signaling in the tumor stromal microenvironment contributes to the development of murine pancreatic MCNs in Ptf1a-Cre-LSL-Kras-elastase-tva mice injected with replication-competent-avian-sarcoma (RCAS)-WNT1 viruses [37]. In the present study, we further crossed pancreatic-specific LKB1-deficient mice with conditional β-catenin loss to observe the eradication of cystadenoma. We observed normal pancreas development and function in Pdx-1Cre LKB1^L/L^ β-catenin^L/L^ mice compared to the Pdx-1Cre LKB1^L/L^ MCN model. Later we employed the Wnt inhibitor compound FH535 to attenuate the development of cystadenocarcinoma in LKB1 mutant mice. Consequently, in our analysis of the mouse cytokine array, CCL1, CCL2, CCL27, CXCL16, LIX (CXCL5), TNF RI and RII, and IL-4 were decreased after treatment of Pdx-1CreLKB1^L/L^ mice with FH535. Among these chemokines and cytokines potentially controlled by Wnt/β-catenin pathway, CCL2 has been reported as a target of the β-catenin/TCF/LEF pathway in breast cancer and plays a vital role in stimulating tumor progression and metastasis [54]. Meanwhile, a recent study demonstrated that the activation of Wnt/Ror2 signaling in mesenchymal stem cells promotes cell proliferation of gastric cancer cells by stimulating the secretion of CXCL16 [55]. In addition, another research study revealed that human mesenchymal stem cells treated with conditioned medium contained Wnt5a protein, which stimulates CXCL5 gene expression [56]. Additional studies on other chemokines or cytokine expression related to Wnt/β-catenin signaling are warranted. Overall, our study suggested the importance of β-catenin involved in LKB1 loss-driven MCN pathogenesis, and the major clinical implications of this study suggest targeting the WNT/β-catenin signaling pathway as a novel strategy for managing MCN with LKB1 inactivation.

## 4. Materials and Methods

### 4.1. Genetically Modified Mice and Mouse Genotyping

Pdx-1Cre, LKB1^Loxp/Loxp^ mice, obtained from the Mouse Models of Human Cancers Consortium (MMHCC) under material transfer agreements, were generously made available by Drs. Andrew M. Lowy and Ron DePinho, respectively. Mice were genotyped as described by the MMHCC PCR protocols for strains 01XL5 and 01XN2. The β-catenin^flox/flox^ mouse was developed in the laboratory of Dr. Tsai at National Yang Ming University, Taiwan, in which the conditional β-catenin loxp/loxp allele was engineered to sustain the Cre-mediated excision of exon 2 to 6. All compound mice were maintained on a mixed 129SV/C57BL/6 background. Animals were maintained at the animal center at the Department of Biological Science, National Sun Yat-Sen University (NSYSU), under specific pathogen-free (SPF) conditions and maintained in strict accordance with the principles and guidelines of the Association for the Assessment and Accreditation of Laboratory Animal Care (AAALC) for the care and use of experimental animals, and approved by the NSYSU Institutional Animal Care and Use Committee (IACUC) (Approval number 10732, 01/08/2016). Furthermore, all surgery and killing practices were performed using isoflurane or avertin to ensure minimal suffering. Pancreatic tissue samples were fixed in 10% buffered formalin overnight; washed with 1× phosphate-buffered saline; and transferred to 70% ethanol before paraffin embedding, sectioning, and hematoxylin and eosin staining.

### 4.2. Immunohistochemistry (IHC) and Immunofluorescence (IF)

Hematoxylin and eosin (H&E) staining followed the standard protocol [57]. For IHC analysis, unstained paraffin slides were baked at 56 °C overnight and deparaffinized in xylene solution twice, before being rehydrated sequentially in 95%, 75%, and 40% ethanol and washed with 1× PBS. Slides were cooked for 20 min in 1× antigen retrieval buffer (H3300, Vector Laboratories, Inc., Burlingame, CA, USA), followed by three rinses with 1× PBS. Slides were quenched with 1% hydrogen peroxide for 10 min before being incubated with blocking solution (4% normal horse serum or goat serum in PBS with 0.1% Triton X). After that, sections were incubated with primary antibody diluted in blocking solution overnight at 4 °C. Primary antibodies for anti-insulin (1:1000 dilution; IS002, Dako, Santa Clara, CA, USA), anti-amylase (1:500 dilution; ab21156, Abcam, Boston, MA, USA), anti-glucagon (1:400 dilution; sc-13091, Santa Cruz Biotechnology, Santa Cruz, CA, USA), anti-CK19 (1:1000 dilution; TROMA-III, Hybridoma Bank, University of Iowa, Iowa City, Iowa. United States), anti-Ki67 (1:500 dilution; ab16667, Abcam, Boston, MA, USA), anti-β-catenin (1:500 dilution; sc-7963, Santa Cruz Biotechnology, Santa Cruz, CA, USA), and anti-active-β-catenin (Anti-ABC) (1:1000 dilution; 05–665, Merck Millipore, Billerica, MA, USA) were purchased from Sigma-Aldrich and Merck Millipore. Slides were then washed with 1× PBS five times and incubated with biotinylated secondary antibody (Vector Laboratories Inc, Burlingame, CA, USA; dilution 1:150) in blocking solution for 1 h at room temperature. After washing five times with 1× PBS, the slides were incubated with a Vectastain Elite ABC kit (Vector Laboratories ABC kit; PK-6100) for 30 min at room temperature. After washing with 1× PBS five times, slides were processed for color reaction with peroxidase treatment with the 3,3′-diaminobenzidine (DAB) substrate kit (Vector Laboratories; SK-4100), washed with tap water, and counterstained with hematoxylin. Stained slides were captured using a Carl Zeiss Axioskop 2 plus microscope (Carl Zeiss, Thornwood, NY, USA) [57]. To determine the labeling score for active β-catenin (ABC) and proliferation marker Ki67, the tumor sections were observed microscopically under high-power magnification (×100), and six different microscopic fields per section were photographed as described previously [58]. The Ki67 proliferation index representative of the whole tumor section was then calculated by dividing the number of Ki67 positive cells by the total number of counted cells. The detailed methodology of the quantitative study has been reported [59].

### 4.3. Glucose Tolerance Test (GTT)

Animals were fasted overnight, and their fasting level of blood glucose was evaluated with ACCU-CHEK Active (Roche Diagnostics, Indianapolis, IN, USA). Mice were then injected intraperitoneally (IP) with 1.5 mg/g of body weight of glucose, and blood glucose levels were measured at the indicated times.

### 4.4. Cell Culture, Reagents, RNA Isolation, and cDNA Synthesis

HEK293T cells, HPDEC, BxPC3, MiaPaCa2, Hs766T, Panc-1 and AsPC-1 human PDAC cells were purchased from the Food Industry Research and Development Institute (FIRDI, Hsinchu, Taiwan, ROC), and were grown as previously described [57,60]. Cycloheximide (CHX) and FH535 were purchased from Sigma (Sigma-Aldrich, St. Louis, MO, USA). For the in vitro experiments, the reagents were dissolved in DMSO and diluted in saline to a final DMSO concentration of ≤0.1%. RNA isolation and cDNA synthesis were conducted as previously described [57,60].

### 4.5. Western Blot and Mouse Cytokine Array Analysis

Cells were harvested in RIPA lysis buffer, and protein concentrations were determined as previously described [61,62]. Approximately 50 µg of protein was loaded and separated by SDS-PAGE, transferred to a PVDF membrane (Millipore, Billerica, MA, USA), and incubated with the following primary antibodies: the primary antibodies for anti-β-catenin (1:500 dilution; sc-7963, Santa Cruz Biotechnology, Santa Cruz, CA, USA) and anti-active-β-catenin (Anti-ABC) (1:1000 dilution; 05–665, Merck Millpore, Billerica, MA, USA); anti-cMYC (1:500 dilution; sc788, Santa Cruz Biotechnology); anti-cyclin D1 (1:500 dilution; sc-8396 Santa Cruz Biotechnology); anti-Survivin (1:500 dilution; sc-17779, Santa Cruz Biotechnology); anti-GAPDH (1:1000 dilution; sc-47724, Santa Cruz Biotechnology); anti-β-actin (1:1000 dilution; #A1978 Sigma-Aldrich, St. Louis, MO, USA). Protein expression levels from Western blots were quantified using Image J. The expression of all proteins was quantified with respect to the expression of β-actin or GAPDH as previously described [8]. For cytokine analysis, the RayBio Mouse Cytokine Array C3 (Cat:AAM-CYT-3–4) was purchased from RayBiotech, Inc., Norcross, GA, USA. Sample preparation and hybridization to the array were performed according to the manufacturer’s instructions.

### 4.6. Real-Time–Quantitative PCR Analysis (RT–qPCR)

Total RNA prepared from samples was used for cDNA synthesis. PCR amplification and the results of the delta CT measurements were previously described in detail. Primer sequences used for real-time qPCR here were as follows: human β-catenin forward, 5′-CACAAGCAGAGTG CTGAAGGTG-3′, and reverse, 5′-GATTCCTGAGAGTCCAAAGACAG-3′, and human GAPDH forward, 5′-GTCTCCTCTGACTTCAACAGCG-3′, and reverse, 5′- ACCACCCTG TTGCTGTAGCCAA-3′. The relative abundance of the targeted gene-specific PCR products was normalized to GAPDH, and the results of the delta CT measurements were previously described in detail [61]. These experiments were independently repeated three times.

### 4.7. Cell Proliferation Assay

For cell growth assays, 2 × 10^4^ cells were seeded in 24-well plates and incubated overnight. Cells were incubated for one to five days before 5 mg/mL MTT (thiazolyl blue tetrazolium bromide) (AMRESCO LLC, Solon, OH, USA) was added to 25 µL in 500 µL RPMI medium (Invitrogen, Carlsbad, CA, USA) and incubated for another 2 h for reaction. Medium was removed and cells were treated with 200 µL DMSO (Sigma, St. Louis, MO, USA) before OD570 reading with a BioTek ELISA reader (Molecular Devices LLC, Sunnyvale, CA, USA) [57,62].

### 4.8. Colony Formation Assay

Fifty thousand cells were suspended in 1 mL of 10% FBS + DMEM medium containing 0.3% agarose and plated in triplicate on a firm 0.6% agarose base in 60 mm tissue culture dishes. After 2–3 weeks, cell growing dishes were washed with phosphate-buffered saline (PBS) and fixed with methanol and 0.1% crystal violet. The cell colonies were manually counted and then photographed [62].

### 4.9. Wound-Healing Assay

Cells were pretreated with 0.02% (0.2 mg/mL) mitomycin C for 2 h and wounded by removing a 300–500 mm wide strip of cells across the well with a standard 200 μL yellow tip. Wounded monolayers were washed twice with phosphate-buffered saline buffer solution to remove non-adherent cells. The cells were cultured in low FBS media and incubated for predetermined times to monitor wound closing. Wound closure was recorded by phase-contrast microscopy, according to previously published protocols [61,62].

### 4.10. Transient Transfections and Luciferase Reporter Assays

The TOP/FOP reporter was kindly provided by Dr. Xi He, Harvard Medical School, Boston, MA, USA. Cells of 60% confluence in 24-well plates were transfected using lipofectamine 2000 (Invitrogen). The TOP/FOP luciferase reporter gene construct (200 ng) and 1 ng of the pRL- SV40 Renilla luciferase construct (for normalization) were co-transfected per well as previously described [38]. Transfected cells were allowed to grow overnight before harvest, cell extracts were prepared 24–48 h after transfection, and the luciferase activity was measured using the Dual-Luciferase Reporter Assay kit (Promega, Madison, WI, USA), according to the manufacturers’ protocol, using Monolight 3010 luminometer (BD biosciences, San Jose, CA, USA) at 570 nM. Expression was calculated as the ratio of arbitrary firefly luciferase units normalized to Renilla luciferase [60]. These experiments were independently repeated three times and each treatment consisted of triplicate samples.

### 4.11. Lentivirus Production and shRNA for Gene Knockdown

The plasmids required for shRNA lentivirus production were purchased from the National RNAi Core Facility, Academia Sinica, Taiwan. The pLKO.1-shRNA vector used for the knockdown of human LKB1 (STK11) was TRCN0000000409. The pLKO.1-shEGFP control plasmid was TRCN00000-72190 (EGFP). Lentivirus production and infection were performed according to a previously described protocol [57,60].

### 4.12. Statistical Analysis

All experiments were repeated at least three times. One representative experiment is shown. RT–qPCR and cell proliferation assays are displayed as one representative experiment of three independent experiments, as the mean ± s.e.m. Data measured on a continuous scale were analyzed using Student’s *t*-test and categorical data were subjected to a x^2^ test. A *p*-value of <0.05 was considered significant [60].

## 5. Conclusions

In summary, our results provide important evidence revealing the association between LKB1 and the Wnt/β-catenin pathway in PC, and we have demonstrated the essential roles of the upregulation of Wnt/β-catenin signaling driven by LKB1 inactivation in promoting MCN formation in vivo and increasing PC cell growth, migration, and invasion in vitro. Moreover, because the inactivation of LKB1 is a potential biomarker to predict postoperative prognosis in PC, β-catenin may be a promising therapeutic target for the treatment of PDAC harboring LKB1 mutations.

## Figures and Tables

**Figure 1 ijms-22-04649-f001:**
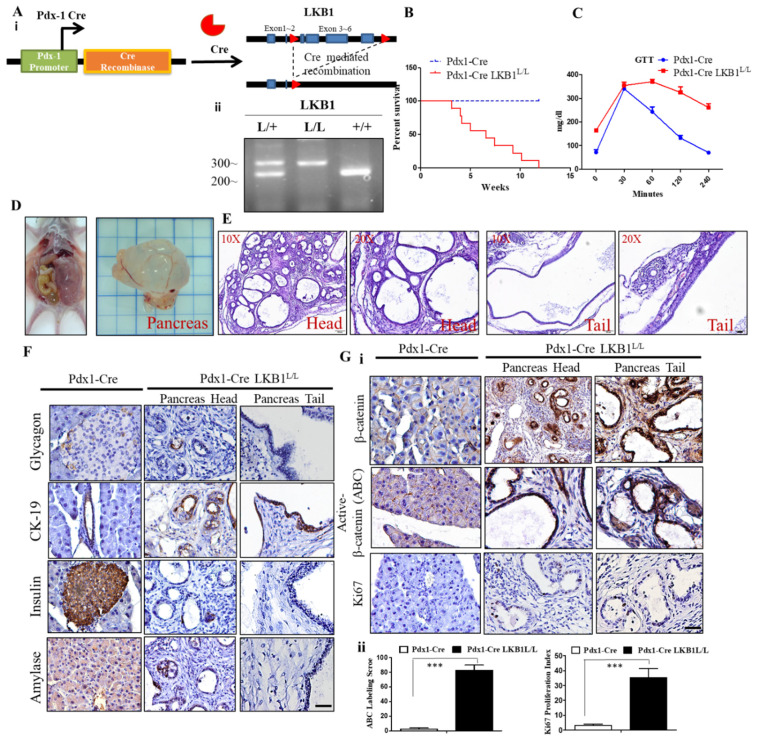
Liver kinase B1 (LKB1) loss induces cystadenoma in the mouse pancreas. (**A**). (**i**) Endogenous allele of LKB1 was conditionally deleted in the pancreas of Pdx1-CreLKB1^L/L^ mice. Structure of the LKB1 floxed allele: Loxp sites were inserted into the introns surrounding exon 3–6. (**ii**)**,** Specific genotyping PCR analysis of the genomic DNA from the tail of the Pdx1-CreLKB1^L/L^, Pdx-1CreLKB1^L/+^, and wild-type mice. (**B**). Kaplan–Meier curve showing the significantly reduced survival time of Pdx1-Cre LKB1^L/L^ mice compared to Pdx1-Cre wild-type mice. (**C**). Glucose tolerance test (GTT) revealed that the fall in blood glucose levels was delayed in Pdx-1Cre LKB1^L/L^ mice compared to that of Pdx-1Cre wild-type mice. (**D**). Autopsy and gross pathology of murine mucin cystadenoma (MCN) lesions in Pdx1-CreLKB1^L/L^ mice at 8 weeks of age. (**E**). Hematoxylin and eosin (H&E) stained sections showing the histology of MCN in the head and tail of the pancreas derived from Pdx1-CreLKB1^L/L^ mice, and histological analysis of the head and tail of the pancreas from Pdx1-CreLKB1^L/L^ mice at magnification x20 and x40, respectively. Scale bar is 100 µm. (**F**). Immunohistochemistry (IHC) analysis detected the positive staining for CK19, whereas there was a loss of insulin, glucagon, and amylase staining in MCN lesions compared to normal pancreata. Scale bar is 100 µm. (**G**). *(***i**) IHC staining revealed increased Ki67 and total and nuclear β-catenin protein levels in Pdx1-CreLKB1^L/L^ mice compared to Pdx-1Cre wild-type controls. Scale bar is 100 µm. (**ii**), The active β-catenin (ABC) labeling score and Ki67 positive cells were quantified per field. Values given are the means ±SE from six random sections for each sample. *** *p* < 0.001.

**Figure 2 ijms-22-04649-f002:**
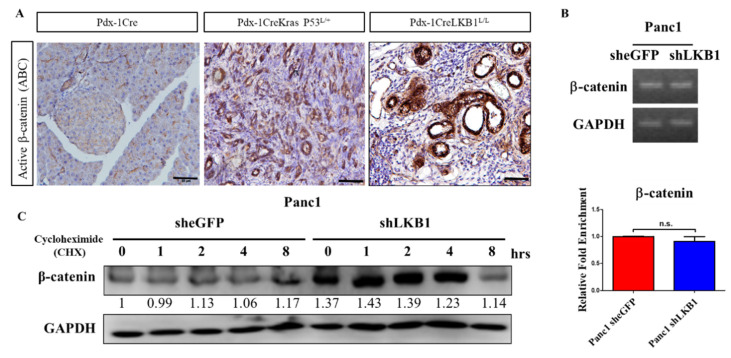
LKB1 loss stimulates the Wnt/β-catenin pathway and enhances the stability of β-catenin in pancreatic cancer (PC). (**A**). Immunostaining demonstrated that active β-catenin (ABC) staining was more intense during pancreatic carcinogenesis. The activation of β-catenin was assessed by immunohistochemistry for ABC in paraffin-embedded pancreas sections of Pdx-1Cre, Pdx-1 Kras P53^L/+^, and Pdx-1CreLKB1^L/L^ mice. Scale bar is 100 µm. (**B**). QRT-PCR analysis of β-catenin in Panc-1 shLKB1 and eGFP control cells. To confirm the mRNA expression levels of β-catenin in Panc-1 shLKB1 and Panc-1 sheGFP control cells, we used reverse real-time PCR. Gel electrophoresis of PCR products by RT-PCR amplification of β-catenin and the GAPDH gene. We found no significant difference of β-catenin gene expression in Panc-1 shLKB1 and sheGFP control cells. The bottom panel shows a quantitative analysis of β-catenin mRNA levels by real-time RT-PCR in Panc-1 shLKB1 and she GFP control cells. n.s. means no statistical significance. (**C**). Western blotting analysis revealed that the knockdown of LKB1 enhanced the stability of β-catenin in Panc-1 cells. Cells were pretreated with cycloheximide (CHX) (50 µM) and incubated for various periods of time and endogenous β-catenin protein expression was detected by Western blot analysis with an anti-β-catenin antibody. GAPDH served as a loading control. Relative densitometric values are provided below the blot images. The results are shown for one of three independent experiments.

**Figure 3 ijms-22-04649-f003:**
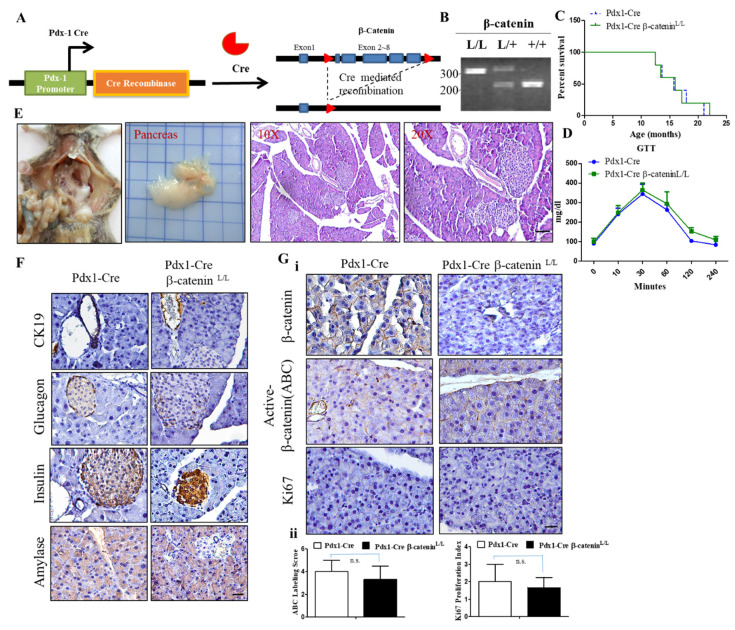
Depletion of β-catenin in the mouse pancreas does not perturb pancreatic development and function. (**A**). Pdx-1Cre-mediated deletion of β-catenin in the pancreas of Pdx1-Cre β-catenin^L/L^ mice. Structure of the β-catenin floxed allele: Loxp sites were inserted into the introns surrounding exon 2 and 6. (**B**). Specific genotyping PCR analysis to detect β-catenin, wild types, and loxp alleles from the tail DNA of the Pdx-1Cre wt, Pdx-1Cre β-catenin^L/+^, and Pdx-1Cre β-catenin^L/L^ offspring; M, 100 bp DNA marker. (**C**). Kaplan–Meier curve showing no difference in the survival time between Pdx-1Cre β-catenin^L/L^ mice and wild-type mice. (**D**). No differences in blood glucose levels were found when blood samples from overnight-fasted Pdx-1Cre β-catenin^L/L^ mice were compared to wild-type mice. Data are presented as means ± SE obtained from six mice/group. (**E**). The gross appearance of the pancreas isolated from Pdx1-Cre β-catenin^L/L^ mice is shown. H&E-stained histological sections of pancreatic tissue from Pdx1-Creβ-catenin^L/L^ mice at both 10X and 20X magnification. Scale bar is 50 µm. (**F**). IHC staining demonstrating that the conditional knockout of β-catenin in the mouse pancreas does not alter insulin, amylase, glucagon, and cytokeratin 19 (CK19) expression in the pancreas in comparison to wild-type mice. Scale bar is 100 µm. (**G**)**.** (**i**) IHC staining showed that the conditional knockout of β-catenin results in the loss of β-catenin protein expression in the mouse pancreas of Pdx-1Cre β-catenin^L/L^ mice. The pancreas of Pdx-1Cre β-catenin^L/L^ mice displayed similar staining patterns with Ki-67 compared to Pdx-1Cre wild-type mice counterparts. Scale bar is 50 µm. (**ii**) The active β-catenin (ABC) labeling score and Ki67 positive cells were quantified per field. Values given are the means ± SE from 6 random sections for each sample. n.s. means no statistical significance.

**Figure 4 ijms-22-04649-f004:**
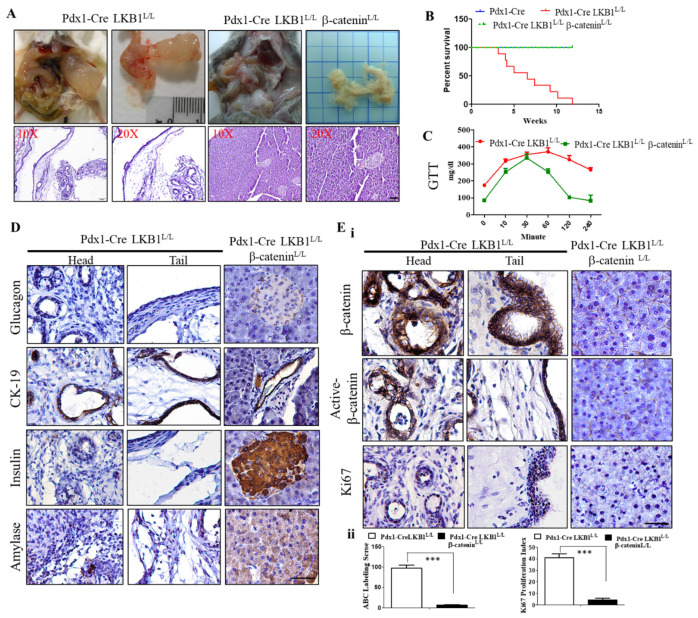
Conditional depletion of β-catenin impairs LKB1 loss-driven MCN formation in mice. (**A**). Gross anatomy of the pancreas and H&E-stained examination of the pancreata of Pdx-1Cre LKB1^L/L^ and Pdx-1Cre LKB1^L/L^ β-catenin^L/L^ mice. The latter displays a completely normal pancreas architecture and function. Scale bar is 100 µm. (**B**). Kaplan–Meier analysis of β-catenin loss significantly extends tumor-free survival in Pdx-1Cre LKB1^L/L^ β-catenin^L/L^ mice. (**C**). Depletion of β-catenin in LKB1 loss-driven cysteadenoma mice restores the normal blood glucose concentration, as determined by a GTT test. *p* < 0.01. (**D**). Immunohistological sections from 20-week-old Pdx1-Cre LKB1^L/L^ and Pdx1-Cre LKB1^L/L^ β-catenin^L/L^ mice stained with antibodies to insulin, glucagon, and amylase and with CK19. Scale bar is 100 µm. (**E**). (**i**) IHC staining for total and active β-catenin and Ki67 in pancreata from Pdx-1CreLKB1^L/L^ mice versus Pdx1-Cre β-catenin^L/L^ LKB1^L/L^ mice. Scale bar is 100 µm. (**ii**) The active β-catenin (ABC) labeling score and Ki67 positive cells were quantified per field. Values given are the means ± SE from 6 random sections for each sample. *** *p* < 0.001.

**Figure 5 ijms-22-04649-f005:**
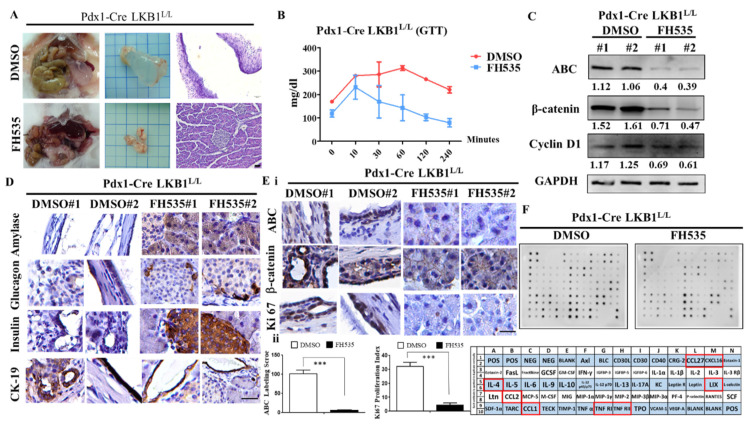
FH535 inhibits pancreatic cystic tumor formation in Pdx-1CreLKB1^L/L^ mice. (**A**). Gross anatomy of the pancreas and H&E stained examination of the pancreata of Pdx-1Cre LKB1^L/L^ mice at 12 weeks of age, with or without FH535 treatment. Scale bar is 100 µm. The bottom panel of the FH535 group exhibits a completely normal pancreas architecture and function. (**B**). Twelve-week-old Pdx-1CreLKB1^L/L^ mice received FH535 treatment that restored the blood glucose concentration to normal (<100 mg/dL) after 2 h, as determined by a GTT test. *p* < 0.01. (**C**). Western blotting analysis revealed that the treatment of FH535 in Pdx-1Cre LKB1^L/L^ mice results in decreasing protein levels of total β-catenin, ABC, and Cyclin D1. GAPDH served as a loading control. Relative densitometric values are provided below the blot images. (**D**). Immunohistological sections from 13-week-old Pdx1-Cre LKB1^L/L^ and Pdx1-Cre LKB1^L/L^ β-catenin^L/L^ mice stained with antibodies against insulin, glucagon, and amylase and CK19. Scale bar is 100 µm. (**E**). (**i**) IHC staining for β-catenin and Ki67 in pancreata from Pdx-1CreLKB1^L/L^ mice versus Pdx1-Cre β-catenin^L/L^ LKB1^L/L^ mice. Scale bar is 100 µm. (**ii**) active β-catenin (ABC) labeling score and Ki67 positive cells were quantified per field. Values given are the means ± SE from 6 random sections for each group. *** *p* < 0.001. (**F**). Mouse cytokine array analysis of pancreas tissue lysate isolated from Pdx-1CreLKB1^L/L^ mice versus Pdx1-Cre β-catenin^L/L^ LKB1^L/L^ mice. Template alignment of the mouse cytokines in the array represented below. The significantly reduced levels of cytokine/chemokine after FH535 treatment are highlighted with red boxes. POS: positive control; NEG: negative control.

**Figure 6 ijms-22-04649-f006:**
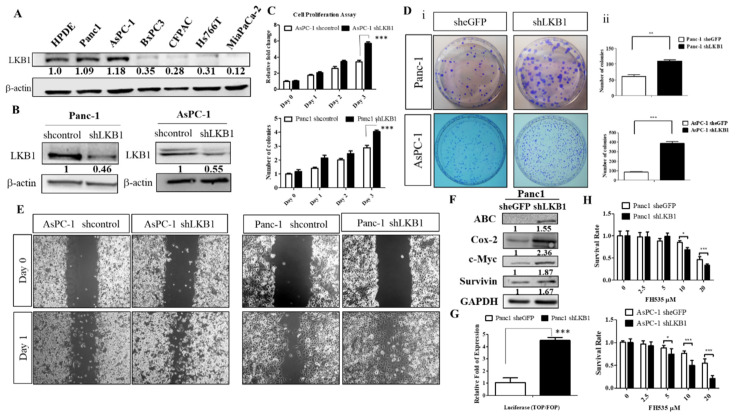
Knockdown of LKB1 promotes cell proliferation and migration/invasion and induces the Wnt/β-catenin signaling pathway in human pancreatic ductal adenocarcinoma (PDAC) cells. (**A**). Western blot analysis of LKB1 protein expression in normal human pancreatic duct epithelial (HPDE) and different human PDAC cell lines. Relative protein expression based on densitometry are listed below the images. (**B**). Western blotting analysis revealed the knockdown efficiency of LKB1 in Panc-1 and AsPC-1 human PDAC cells. The level of knockdown was quantified by densitometric analysis of Western blots using ImageJ software. (**C**). The MTT (thiazolyl blue tetrazolium bromide) cell proliferation assay showed that the knockdown of LKB1 promotes the human PDAC cell proliferation rate in vitro. *** *p* < 0.001. (**D**). (**i**) The sphere formation assay determined that the knockdown of LKB1 significantly increases colony formation in human PDAC cells. (**ii**) Quantitation of cell colony numbers of Panc-1 shLKB1, AsPC-1 shLKB1 and their sheGFP control cells. *** *p* < 0.001. (**E**). Knockdown of LKB1 in human Panc-1 and AsPC-1 PDAC cells increases cell motility by using an in vitro wound healing assay. (**F**). Western blotting indicated that LKB1 knockdown in human PDAC cells leads to upregulation of the levels of ABC, Cox-2, c-MYC, and survivin proteins served as downstream effectors of the Wnt pathway. Relative densitometric values are provided below the blot images. (**G**). Luciferase assay showing the increased TOP/FOP luciferase reporter activity in Panc-1 shLKB1 cells, compared with Panc-1 sheGFP control cells. *** *p* < 0.001. (**H**). MTT cell survival assays showed that FH535 significantly reduced cell proliferation rates in Panc-1 shLKB1, AsPC-1shLKB1, and eGFP control cells. * *p* < 0.01; *** *p* < 0.001.

## Data Availability

Not applicable.

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
