# Peer review of "Inhibition of β-Catenin Activity Abolishes LKB1 Loss-Driven Pancreatic Cystadenoma in Mice"

_ijms, 2021, doi:10.3390/ijms22094649_

Round 1
Reviewer 1 Report
In this manuscript, the authors describe the loss of LKB1 in development of murine pancreatic cystadenoma, and the effect of B-catenin loss or inhibition on development of these cystadenomas.
The authors describe what is known about LKB1 in pancreatic ductal adenocarcinoma (PDAC), but what is known about LKB1 mutations or expression in human pancreatic MCNs? It is unclear how physiologically relevant LKB1 loss is to human cystic neoplasms. Furthermore, the reason for including experiments examining LKB1 knockdown in PDAC cell lines is unclear in the last figure, since the rest of the manuscript focuses on MCNs.
The logic behind investigating the Wnt pathway in LKB1 null pancreatic lesions is also unclear- why was a link between these pathways investigated?
The staining intensity of much of the immunohistochemistry performed is difficult to observe, and adding quantification of staining intensity could be helpful.
A Pdx1-Cre control for comparison should also be included in Figure 4C-E, even though similar results are shown in previous figures.
Experimental details of glucose tolerance tests should be included in the methods section. How long were mice treated with FH535 before performing the GTT?
Similarly, what was the experimental endpoint for in vivo experiments with FH535? At different points, the endpoint is referred to as either 6, 14, or 20 weeks. This should be clarified in the test and Figure legends.
Moderate editing of English language is needed.
Minor changes:
The text on Figure 5F (bottom) cannot be read.
Glucose tolerance tests in mice are usually abbreviated “GTT”. In humans, it is an oral glucose tolerance test “OGTT”. (Not GOTT).
Author Response
Comments from Reviewer 1
In this manuscript, the authors describe the loss of LKB1 in development of murine pancreatic cystadenoma, and the effect of b-catenin loss or inhibition on development of these cystadenomas.
Response to Reviewer 1
We thank the reviewer for his/her constructive comments and suggestions to help improve this manuscript. Below, we have addressed all of the points raised by the review#1.
The authors describe what is known about LKB1 in pancreatic ductal adenocarcinoma (PDAC), but what is known about LKB1 mutations or expression in human pancreatic MCNs? It is unclear how physiologically relevant LKB1 loss is to human cystic neoplasms. Furthermore, the reason for including experiments examining LKB1 knockdown in PDAC cell lines is unclear in the last figure, since the rest of the manuscript focuses on MCNs.
Reply: Mucinous cystic neoplasms (MCN) of the pancreas are rare tumors, constituting 2–5% of pancreatic neoplasms1. The molecular changes underlying MCN formation and progression are not entirely clear. KRAS and PIK3CA mutations have been detected in MCNs with low-grade dysplasia, while mutations of P53, p16, and SMAD4 have been mainly observed in high-grade dysplasia and invasive carcinomas1,2. Somatic mutations of the STK11/LKB1 genes are rarely seen in NCNs, however, similar to PIK3CA mutations, mutations of LKB1 affect PI3K/Akt/mTOR signaling to exert its oncogenic potentials. Over 70% of patients with Peutz–Jeghers syndrome (PJS) have been detected with a pathogenic mutation in LKB1 gene. Importantly, patients with PJS (germline mutation in LKB1) are also at increased risk for intraductal papillary mucinous neoplasms (IPMNs) and pancreatic cysts3. Notably, the conventional knockout LKB1 (+/−) mice not only develop to gastrointestinal polyps of which the histological characteristics resemble those of the Peutz-Jeghers syndrome hamartomas but also more than 70 % male mice develop hepatocellular carcinomas (HCCs) within 50 weeks, and individuals with Peutz-Jeghers have a 130-fold increased risk of pancreatic cancer4. Several lines of evidence have shown that low expression of LKB1 is associated with significant worse over survival as compared with those patients with LKB1 high expression tumors in human PDAC. Although inactivation of LKB1 has been reported only in 4-7% of sporadic PDAC samples, low LKB1 expression level actually occurs in 20-25% of human PDAC and negatively impacts the clinical outcomes of PDAC patients5.
Meanwhile, in the present study, we also apply the in vitro human PDAC cell line models to investigate whether LKB1 knockdown affects human PDAC tumorigenic properties. Our in vitro data provided important evidences to confirm that the essential roles for the upregulation of Wnt/b-catenin signaling driven by LKB1 inactivation in the pancreatic and inactivation of LKB1 leads to increased growth, migration and invasion of pancreatic cancer cells in vitro.
References:
- Distler M. , Aust D. , Weitz J., C Pilarsky C., Robert G, et al.,Precursor lesions for sporadic pancreatic cancer: PanIN, IPMN, and MCN. Biomed Res Int. 2014;2014: 474905 2014 Mar 24.
- Michelle Dd , Burcu S, Serdar B, Andrew S G, N Volkan A et al.,Molecular genetics of pancreatic neoplasms and their morphologic correlates: an update on recent advances and potential diagnostic applications. Am J Clin Pathol. 2014 Feb; 141(2):168-80.
- Su GH, Hruban RH, Bova GS, et al. Germline and somatic mutations of the STK11/LKB1 Peutz-Jeghers gene in pancreatic and biliary cancers. Am J Pathol. 1999;154(6):1835–1840.
- Nakau, M., et al., Hepatocellular carcinoma caused by loss of heterozygosity in Lkb1 gene knockout mice. Cancer research, 2002. 62(16): p. 4549-4553.
- Yang, J.Y., et al., Decreased LKB1 predicts poor prognosis in Pancreatic Ductal Adenocarcinoma. Sci Rep, 2015. 5: p. 10575.
The logic behind investigating the Wnt pathway in LKB1 null pancreatic lesions is also unclear- why was a link between these pathways investigated?
Reply: Our previous study published in 2015 Molecular Caner Research demonstrated that LKB1 regulates GSK3b phosphorylation through LKB1-APC-GSK3b interaction and to influence the activity of Wnt signaling pathway in lung cancer. Therefore, we hypothesized that LKB1 deficiency driven MCN formation in the mouse pancreas may also be associated with the stabilize/ active b-catenin protein resulting in the activation of Wnt pathway to involve in the induction of pancreatic MCN in mice.
The staining intensity of much of the immunohistochemistry performed is difficult to observe, and adding quantification of staining intensity could be helpful.
Reply: We have increased the resolution of IHC images in the revised manuscript.
A Pdx1-Cre control for comparison should also be included in Figure 4C-E, even though similar results are shown in previous figures.
Reply: We have already shown the IHC staining patterns in the pancreas of Pdx-1Cre control mice in Figure 1 and Figure 3. in our revised manuscript.
Experimental details of glucose tolerance tests should be included in the methods section. How long were mice treated with FH535 before performing the GTT?
Reply: Done. We have added the experimental details of glucose tolerance tests in the methods section of our revised manuscript.
Similarly, what was the experimental endpoint for in vivo experiments with FH535? At different points, the endpoint is referred to as either 6, 14, or 20 weeks. This should be clarified in the test and Figure legends.
Reply: This point is well-taken. We have clarified it in the Figure legend.
Moderate editing of English language is needed.
Reply: This revised manuscript has gone through intensive English editing to correct some typos, mislabeled data in this manuscript and we have rewritten the figure legends, did some grammar checking and rephrased the discussion section to make our manuscript suitable for publication in this journal.
Minor changes:
The text on Figure 5F (bottom) cannot be read.
Reply: We thank the reviewer for the constructive comments and suggestions. The point is well taken. We made a new Excel spreadsheet into a table to replace it.
Glucose tolerance tests in mice are usually abbreviated “GTT”. In humans, it is an oral glucose tolerance test “OGTT”. (Not GOTT).
Reply: We thank the reviewer for the suggestion. We have changed it to GTT in our revised manuscript
Reviewer 2 Report
The manuscript entitled: “Inhibition of β-catenin activity abolishes LKB1 loss driven pancreatic cystadenoma in mice”, by Hsieh MJ, Weng CC and Lin YC, et al., is a study in which the authors correlate the development of pancreatic lesions corresponding to cystadenomas with deficient expression the tumor suppressor LKB1 using a novel murine models.
The authors engineered a Pdx-1Cre LKB1L/L mouse model to study how the lack of LKB1 expression in pancreatic epithelium leads to formation of cystadenomas, which eventually could progress to pancreatic cancer. As part of this murine model characterization, the authors showed evidence of excessive β-catenin expression in the epithelial cells within these cystic structures, correlating also with high levels of active β-catenin, both of which seem to be associated with a shorter lifespan of these animals. The authors prevented these LKB1-loss induced pancreatic alterations by either genetic ablation of β-catenin, or by chemical silencing of the Wnt/β-catenin signaling pathway, via small molecule inhibitor FH535 in a LKB1-null background.
In general, this study attempts to understand the role of LKB1 in the process of transformation from normal pancreatic epithelium to pre-malignant cystadenoma lesions, for which they suggest β-catenin is an important player. For that, the authors relay on the characterization of their newly generated pancreatic precancerous model, basing their conclusions on the expression profile of some related biomarkers, supported by animal survival studies, and pancreas functional assays. Additionally, the authors conducted in vitro functional assays using well-known pancreatic cancer cells subjected to a reduced expression of LKB1.
After evaluating several aspects of the current study, this reviewer finds this work weak, since although the authors have a powerful pre-malignant pancreatic model in their hands, this to have been sub-utilized, channeling their efforts to the characterization of their model, rather than exploding its power to better understand the mechanisms behind the role of LKB1 on the canonical and/or non-canonical signaling pathways of Wnt/β-catenin, either during the onset of pancreatic cystadenoma or progression to malignant stages.
Author Response
For Reviewer 2
The authors engineered a Pdx-1Cre LKB1L/L mouse model to study how the lack of LKB1 expression in pancreatic epithelium leads to formation of cystadenomas, which eventually could progress to pancreatic cancer. As part of this murine model characterization, the authors showed evidence of excessive β-catenin expression in the epithelial cells within these cystic structures, correlating also with high levels of active β-catenin, both of which seem to be associated with a shorter lifespan of these animals. The authors prevented these LKB1-loss induced pancreatic alterations by either genetic ablation of β-catenin, or by chemical silencing of the Wnt/β-catenin signaling pathway, via small molecule inhibitor FH535 in a LKB1-null background.
In general, this study attempts to understand the role of LKB1 in the process of transformation from normal pancreatic epithelium to pre-malignant cystadenoma lesions, for which they suggest β-catenin is an important player. For that, the authors relay on the characterization of their newly generated pancreatic precancerous model, basing their conclusions on the expression profile of some related biomarkers, supported by animal survival studies, and pancreas functional assays. Additionally, the authors conducted in vitro functional assays using well-known pancreatic cancer cells subjected to a reduced expression of LKB1.
After evaluating several aspects of the current study, this reviewer finds this work weak, since although the authors have a powerful pre-malignant pancreatic model in their hands, this to have been sub-utilized, channeling their efforts to the characterization of their model, rather than exploding its power to better understand the mechanisms behind the role of LKB1 on the canonical and/or non-canonical signaling pathways of Wnt/β-catenin, either during the onset of pancreatic cystadenoma or progression to malignant stages.
Response to Reviewer 2
We would like to thank you for your time and valuable comments that helped to improve the manuscript substantially. In this study, our data provided important evidences revealing the association between LKB1 and the Wnt/b-catenin pathway in pancreatic cancer, and we demonstrated the essential roles for the upregulation of Wnt/b-catenin signaling driven by LKB1 inactivation in promoting MCN formation in vivo and increasing pancreatic cancer cell growth, migration and invasion in vitro. Importantly, because inactivation of LKB1 is a potential biomarker to predict postoperative prognosis in pancreatic cancer, b-catenin may be a promising therapeutic target for the treatment of PDAC harboring LKB1 mutations.
Reply: In line 152-154 of page 5, we added the following sentence in the revised version.
“Additional work is required to elucidate the detailed mechanisms behind the role of LKB1 on the canonical and/or non-canonical Wnt signaling pathways, either during the onset of pancreatic cystadenoma or progression to invasive cystadenocarcinoma”.

Round 2
Reviewer 1 Report
It appears that very little was changed in this manuscript in response to reviewer concerns. Additional revisions in the form of added or revised data, re-written text, English editing, for example, are needed to address the reviewer concerns.
Author Response
Dear Reviewer 1:
We highly appreciate the reviewers’ insightful and helpful comments on our manuscript. In this study, our data provided important evidences revealing the association between LKB1 and the Wnt/β-catenin pathway in pancreatic cancer, and we demonstrated the essential roles for the upregulation of Wnt/β-catenin signaling driven by LKB1 inactivation in promoting MCN formation in vivo and increasing pancreatic cancer cell growth, migration and invasion in vitro. Importantly, because inactivation of LKB1 is a potential biomarker to predict postoperative prognosis in pancreatic cancer, β-catenin may be a promising therapeutic target for the treatment of PDAC harboring LKB1 mutations.
Following the reviewer’s comments, we have requested for English Editing Service from MDPI website to perform grammar checking, data revision, re-written text and English editing etc.. Meanwhile, we added more experimental data (Figure 6A), substantially improved the quality of all IHC images, quantitated active β-catenin (ABC) score and Ki67 proliferation index in Figure1G, 3G, 4E and 5E, semi-quantified semi-quantitative analysis of our western blots in Figure 2C, 5C, 6A and 6F, rewritten the figure legends and rephrased the results and discussion sections in this revised manuscript. We believe that our manuscript has been considerably improved as a result of these revisions, and hope that our revised manuscript is acceptable for publication in IJMS.
We would like to thank you once again for your consideration of our work and inviting us to submit the revised manuscript. We look forward to hearing from you.
Best regards,
Kuang
Kuang-hung Cheng, Ph.D
Professor of Biomedical Science Institute
National Sun Yat-Sen University
Kaohsiung, Taiwan 807
TEL: 886-7-5252000 ext 5817

Reviewer 2 Report
The authors have added an important statement regarding the need to dissect the role of LKB1 on the canonical and/or non-canonical Wnt signaling pathways during the malignization process from pancreatic cystadenoma to invasive cystadenocarcinoma. Additionally, the authors have added important information to their data discussion, and the materials and methods section.
This reviewer acknowledges these additions, however, no attempt of providing additional information regarding a novel mechanism explaining how the loss of LKB1 in pancreatic cystadenoma, which correlates with an increase of active β-catenin, and later is associated with pancreatic cancer (PC) progression was done and results frustrating that several downfalls are still present.
It is worth saying that the study presented could be an important piece of information, especially pointing to the generated premalignant model of PC, which offers an interesting system to study the progression of this disease. Nevertheless, since the present form of this manuscript lacks enough details, it is very difficult to take it as a disease murine model publication or as a discovery study, for which the material revised does not offer enough scientific rigor. This concern is based on the fact that the study of LKB1 in GI cancers has already offered information regarding the role of this tumor suppressor gene in relationship with β-catenin (e. g. Ma LG, et al. 2016. Int J Mol Med. DOI: 10.3892/ijmm.2016.2494). In addition, recent work has been done on pancreatic cancer, elucidating more details on how Kras mutations synergize with LKB1 inactivation, lead to premalignant (IPMN) modifications of the pancreatic epithelium. (Collet L, et al. Gut. 2020. DOI: 10.1136/gutjnl-2018-318059).
Therefore, in order to substantially improve the presented study, several major points need to be addressed:
- Provide information regarding a mechanism explaining how the loss of LKB1 prevents degradation of β-catenin and how this leads to its enhanced activation in the pre-malignant pancreatic epithelium.
- Support the in vitro results by evaluating if genetically silencing of LKB1 in a disease-relevant cell line, could recapitulate the observations described in the Pdx-1Cre LKB1L/L mouse murine model when implanted in mice. Additionally, test if the double ablation of LKB1 + β-catenin prevents the formation of the previously mentioned pancreatic lesions.
- For the introduction as well as for the discussion sections, include relevant information from references provided lines above, to support or contrast the data presented in this work.
- The authors refer to a cytokine expression profile conducted in the initial characterization of their model (line 99). These data need to be presented, described, and discussed, especially since later in the study seems to be relevant when the loss of LKB1 is overcome by the β-catenin chemical inhibitor.
- Regarding the murine model characterization presented in figure 1, comparative H&E stain, as well as IHC of glycagon, CK-19, insulin, amylase, and β-catenin in human MCN tissue must be provided.
- From figure 1G and ahead, intensity and cellular localization of markers presented (β-catenin and ABC) in all figures, must be scored, results graphed, and described in the corresponding results section.
- Also for figure 1G, authors are encouraged to describe the β-catenin positive stain in the stroma compartment and discuss this observation under the light of the Wnt signaling pathway potential crosstalk between tumor and stroma.
- Results in figure 2C must be complemented by the assessment of quantitated levels of active beta-catenin (nuclear localization) in PANC1 shLKB1, either by WB of cell-fractionated lysates or by IF images analysis (data must be normalized to cellularity).
- Results description for figure 5C appears to be in discordance with the presented levels of Cyclin D1 in the membrane; authors must verify this result. Also, the OD values of all markers must be presented as normalized ratios to GAPDH OD.
- Results in figure 5F cannot be read. This result should be contrasted with the cytokine profile requested (and mentioned in the results) for figure 1.
- For the in vitro assays where pancreatic cancer cell lines were used, the authors must explain why Panc-1 and AsPC-1 cells were chosen as representative cell lines within the context of their MCN model.
- In general, extensive improvement needs to be done for figure 6. Graphs in section B need to reflect statistical analysis; results in section C needs to be complemented with quantitative data, similar quantitation is needed for results in section D. Results presented in section E doesn't seem to reflect the description in the figure legend. Moreover, if the authors intended to prove that Panc-1 cells expressing decreased levels of LKB1 have more intrinsic Wnt signaling pathway-related transcriptional activity, they should provide data from a positive control triggering the pathway.
- If the authors have additional data regarding p-AMPKα and C-Myc related to LKB1 within the context of the present study, these must be provided or else remove any statement related that could be misleading.
- Lastly, the authors are urged to discuss data obtained regarding the cytokines expression profiles of pre-malignant and malignant cells lacking LKB1, under the light of MCN progression to pancreatic cancer and the possible impact of those on the altered Wnt signaling pathway observed in their model, and within the context of the human disease.
Additional minor points are listed for improvement:
- Since in most cases it is difficult to appreciate details in the tissue microphotographs presented, the authors must improve substantially the quality of their images according to the journal standards.
- Figure 2A legend is missing scale bar size information.
- In line 195, "(Figure 3F)" seems to be redundant since the lines seem to be introducing the results described in lines 196-197 and referred to (Figure 3F and 3G).
- Title in results section 2.6 should read: shRNA knockdown of LKB1 expression increases cell proliferation and migration of PDAC cells in vitro.
- It appears that the authors missed including the results observed from AsPC-1 cells in the results described in figure 6D (lines 319-320).
- In section 4.2 Immunohistochemistry (IHC) and immunofluorescence (IF), the authors should provide the antibody concentration or dilution ratio used.
Author Response
From Reviewer 2 Comments and Suggestions
The authors have added an important statement regarding the need to dissect the role of LKB1 on the canonical and/or non-canonical Wnt signaling pathways during the malignization process from pancreatic cystadenoma to invasive cystadenocarcinoma. Additionally, the authors have added important information to their data discussion, and the materials and methods section.
This reviewer acknowledges these additions, however, no attempt of providing additional information regarding a novel mechanism explaining how the loss of LKB1 in pancreatic cystadenoma, which correlates with an increase of active β-catenin, and later is associated with pancreatic cancer (PC) progression was done and results frustrating that several downfalls are still present.
It is worth saying that the study presented could be an important piece of information, especially pointing to the generated premalignant model of PC, which offers an interesting system to study the progression of this disease. Nevertheless, since the present form of this manuscript lacks enough details, it is very difficult to take it as a disease murine model publication or as a discovery study, for which the material revised does not offer enough scientific rigor. This concern is based on the fact that the study of LKB1 in GI cancers has already offered information regarding the role of this tumor suppressor gene in relationship with β-catenin (e. g. Ma LG, et al. 2016. Int J Mol Med. DOI: 10.3892/ijmm.2016.2494). In addition, recent work has been done on pancreatic cancer, elucidating more details on how Kras mutations synergize with LKB1 inactivation, lead to premalignant (IPMN) modifications of the pancreatic epithelium. (Collet L, et al. Gut. 2020. DOI: 10.1136/gutjnl-2018-318059).
Dear Reviewer 2:
Thank you very much for having considered our manuscript. We are very happy to have received a positive evaluation, and we would like to express our appreciation to you and both Reviewers for the thoughtful comments and helpful suggestions. Reviewer #2 raised several concerns, which we have carefully considered and made every effort to address. We fundamentally agree with all the comments made by the Reviewers, and we have incorporated corresponding revisions into the manuscript (version R2). Our detailed, point-by-point responses to the editorial and reviewer comments are given below, whereas the corresponding revisions are marked in colored text in the manuscript file (version R2). In addition, we have requested for English Editing Service from MDPI website to perform grammar checking, data revision, re-written text and English editing etc. Meanwhile, we added more experimental data (Figure 6A), substantially improved the quality of all IHC images, quantitated active β-catenin (ABC) score and Ki67 proliferation index in Figure1G, 3G, 4E and 5E, semi-quantified semi-quantitative analysis of our western blots in Figure 2C, 5C, 6A,6B and 6F, rewritten the figure legends and rephrased the results and discussion sections in this revised manuscript. We believe that our manuscript has been considerably improved as a result of these revisions, and hope that our revised manuscript is acceptable for publication in IJMS.
Reply: Thanks for the reviewer’s comment. We included this reference (Collet L, et al. Gut. 2020. DOI: 10.1136/gutjnl-2018-318059) in our discussion section (On page 13, line 423-427). Meanwhile, we added the description of “Concurrently, Louis and his colleagues reported that Kras mutations synergize with LKB1 inactivation in the pancreas, leading to the development of IPMN in mice. However, based on their findings, they concluded that the lack of β-catenin did not impede formation of intraductal papillae and their progression to papillary lesions in IPMN, which probably because the activating KrasG12D mutation combined with LKB1 ablation produced more synergistic effects in promoting development of IPMN [53]. “
Therefore, in order to substantially improve the presented study, several major points need to be addressed:
- Provide information regarding a mechanism explaining how the loss of LKB1 prevents degradation of β-catenin and how this leads to its enhanced activation in the pre-malignant pancreatic epithelium.
Reply: (In page 5, line 161-163 of the revised manuscript) We described that “ Our data supported the hypothesis that LKB1 deficiency leads to a stabilized/active b-catenin protein to prevent b-catenin degradation, which results in decreasing b-catenin degradation and maintains the activation of Wnt signaling in PDAC cells. Intriguingly, our previous study demonstrated that LKB1 regulates GSK3β phosphorylation through the LKB1–APC–GSK3b interaction and influences the activity of the Wnt signaling pathway in lung cancer. Additional work is required to further elucidate the detailed mechanisms underlying the role of LKB1 in canonical and/or non-canonical Wnt signaling pathways, either during the onset of pancreatic cystadenoma or progression to metastatic malignancy.
- Support the in vitro results by evaluating if genetically silencing of LKB1 in a disease-relevant cell line, could recapitulate the observations described in the Pdx-1Cre LKB1L/Lmouse murine model when implanted in mice. Additionally, test if the double ablation of LKB1 + β-catenin prevents the formation of the previously mentioned pancreatic lesions.
Reply: We understand the reviewer's viewpoint here. In the present study, we later decided to use human PDAC cell lines for in vitro functional tests since primary cell culture didn’t work (very tough, almost impossible according to our experience) for the establishment of primary cell lines form Pdx-1Cre LKB1L/L cystic precancerous or benign tumor model.
For the introduction as well as for the discussion sections, include relevant information from references provided lines above, to support or contrast the data presented in this work.
Reply: Thanks for the reviewer’s suggestion. In the result section ((In page 5, line 161-163 ), and the results section (page 13, line 434-437) we added relevant references to support our findings here, and we also explained some contrast the data published by other groups in the discussion section ( page 13, line 423-426) .
- The authors refer to a cytokine expression profile conducted in the initial characterization of their model (line 99). These data need to be presented, described, and discussed, especially since later in the study seems to be relevant when the loss of LKB1 is overcome by the β-catenin chemical inhibitor.
Reply: Thank you for reviewer suggestion. To prevent any misleading, we removed the sentence which described about the cytokine expression profile conducted in the initial characterization of this model, since it would need more detailed explanation. We just keep the later one which was conducted to be relevant when the loss of LKB1 is overcome by the β-catenin chemical inhibitor in Figure 5F.
- Regarding the murine model characterization presented in figure 1, comparative H&E stain, as well as IHC of glycagon, CK-19, insulin, amylase, and β-catenin in human MCN tissue must be provided.
Reply: Thanks for the reviewer’s constructive suggestion. We added some description about β-catenin in human MCN tissue about have included the references in the results section of this revised manuscript.
In page 3; line 116-118; we added” One recent work by Makoto and his colleagues suggested that activation of Wnt signaling within the stroma might contribute to development of human pancreatic mucinous cystic neoplasms [37]. Therefore, we explored the link between loss of LKB1 and activation of Wnt signaling in our Pdx-Cre-driven LKB1 loss mouse model of MCN.” In our revised manuscript.
- From figure 1G and ahead, intensity and cellular localization of markers presented (β-catenin and ABC) in all figures, must be scored, results graphed, and described in the corresponding results section.
Reply: we added (β-catenin and ABC) staining score and Ki67 proliferation index in the revised version.
- Also for figure 1G, authors are encouraged to describe the β-catenin positive stain in the stroma compartment and discuss this observation under the light of the Wnt signaling pathway potential crosstalk between tumor and stroma.
Reply: OK, we added the reference to discuss the β-catenin positive nuclear stain in the stroma compartment of our murine MCNs in the result section of this revised manuscript. On page 3, line 116-124; of the result section, we added” One recent work by Makoto and his colleagues suggested that activation of Wnt signaling within the stroma might contribute to development of human pancreatic mucinous cystic neoplasms [37]. Therefore, we explored the link between loss of LKB1 and activation of Wnt signaling in our Pdx-Cre-driven LKB1 loss mouse model of MCN. Here we identified that the mouse MCN lesions stained positively for Wnt signaling pathways. These cystic lesions exhibited an increased expression of Ki67 and total and active β- catenin activities compared to the surrounding normal pancreas, as shown by IHC analysis (Figure 1Gi-ii). Of note, our findings confirmed that that the active β-catenin (ABC) positive nuclear staining can occur within the cystic epithelial tumors and stroma cells, which…..
- Results in figure 2C must be complemented by the assessment of quantitated levels of active beta-catenin (nuclear localization) in PANC1 shLKB1, either by WB of cell-fractionated lysates or by IF images analysis (data must be normalized to cellularity).
Reply: Thank you for the suggestion and careful review. In Figure 6F, we have already shown the reduction of active b catenin (ABC) levels (using specific targeted Dephospho β-catenin (clone 8E7) antibody) in Panc1 shLKB1 cells.
- Results description for figure 5C appears to be in discordance with the presented levels of Cyclin D1 in the membrane; authors must verify this result. Also, the OD values of all markers must be presented as normalized ratios to GAPDH OD.
Reply: This point is well taken. We added semi quantitative analysis data from our western blots
- Results in figure 5F cannot be read. This result should be contrasted with the cytokine profile requested (and mentioned in the results) for figure 1.
Reply: We thank the reviewer for the constructive comments and suggestions. The point is well taken. We made a new Excel spreadsheet into a table and enlarge the font size to increase image resolution.
- For the in vitro assays where pancreatic cancer cell lines were used, the authors must explain why Panc-1 and AsPC-1 cells were chosen as representative cell lines within the context of their MCN model.
Reply: This point is well taken. We added a new experimental data in Figure 6A, which we showed that we first screened the LKB1 expression levels in a variety of human PDAC cell lines to explain why Panc-1 and AsPC-1 cells were chosen as representative cell lines to knockdown LKB1 in pancreatic cancer in this article.
- In general, extensive improvement needs to be done for figure 6. Graphs in section B need to reflect statistical analysis; results in section C needs to be complemented with quantitative data, similar quantitation is needed for results in section D. Results presented in section E doesn't seem to reflect the description in the figure legend. Moreover, if the authors intended to prove that Panc-1 cells expressing decreased levels of LKB1 have more intrinsic Wnt signaling pathway-related transcriptional activity, they should provide data from a positive control triggering the pathway.
Reply: All points are well taken. We have provided additional information for Figure 6 according to the reviewer’s suggestion. Meanwhile, as shown in Figure 6 F, we already have shown the increased TOP/FOP luciferase activity in Panc-1 shLKB1 as compared to control cells to refer more intrinsic Wnt signaling pathway in LKB1 knockdown cells. Meanwhile, our previous publication has already shown that an increased TOP/FOP luciferase activity after administration of lithium chloride (10 mmol/L LiCl), the Wnt/b-catenin activator to serve as the positive control. We added this reference in the Materials and Methods section of this revised manuscript.
- If the authors have additional data regarding p-AMPKα and C-Myc related to LKB1 within the context of the present study, these must be provided or else remove any statement related that could be misleading.
Reply: We apologize for any confusion, and we have rephrased the results and removed any misleading statement which we didn’t not present any result to support it
- Lastly, the authors are urged to discuss data obtained regarding the cytokines expression profiles of pre-malignant and malignant cells lacking LKB1, under the light of MCN progression to pancreatic cancer and the possible impact of those on the altered Wnt signaling pathway observed in their model, and within the context of the human disease.
Reply: We thank the reviewer for highlighting this point. We added some sentences in the results section (In page 19, line 311-314). We tend to investigate the potential roles of these cytokines induced by LKB1 loss during the pathogenesis of pancreatic MCNs in our further manuscripts.
In page 19, line 311-314; we added” Thus, there are clearly many chemokines and chemokines that are upregulated by LKB1 loss in the pancreas, and are relevant to the pathogenesis of pancreatic MCNs. Most need further investigation to confirm their specific impact on MCN pathology and clinical features, and whether they may be targeted therapeutically using in vivo models.”
Additional minor points are listed for improvement:
- Since in most cases it is difficult to appreciate details in the tissue microphotographs presented, the authors must improve substantially the quality of their images according to the journal standards.
Reply: Done, we have improved improve substantially the quality of all our IHC images in this revised version.
- Figure 2A legend is missing scale bar size information.
Reply: Done. We added the information of the scale bar size in the Figure 2A legend.
- In line 195, "(Figure 3F)" seems to be redundant since the lines seem to be introducing the results described in lines 196-197 and referred to (Figure 3F and 3G).
Reply, Ok. We apologize for any confusion. We have removed the redundant statement in line 195 (refer to line 203-204 of the revised version).
- Title in results section 2.6 should read: shRNA knockdown of LKB1 expression increases cell proliferation and migration of PDAC cells in vitro.
Reply: Thank you for the reviewer carefully review. Done. This point is well taken.
- It appears that the authors missed including the results observed from AsPC-1 cells in the results described in figure 6D (lines 319-320).
Reply: OK, we included the results observed from AsPC-1 cells in the revised manuscript.
- In section 4.2 Immunohistochemistry (IHC) and immunofluorescence (IF), the authors should provide the antibody concentration or dilution ratio used.
Reply: Done. We added the dilution conditions for IHC and western blot methods in the Materials and Methods section.
Thanking you once again for your kind consideration of our revised and improved this manuscript for publication.
Sincerely yours
Kuang
Kuang-hung Cheng, Ph.D
Professor of Biomedical Science Institute
National Sun Yat-Sen University
Kaohsiung, Taiwan 807
TEL: 886-7-5252000 ext 5817
Fax: 886-7-5250197
Email: khcheng@faculty.nsysu.edu.tw

Round 3
Reviewer 1 Report
This manuscript has been revised and the authors have responded to suggested edits.
Author Response
Response: We thank the reviewer for the compliments and especially for the constructive suggestions before. The reviewer has no any further comment on our revised manuscript since this revised manuscript has been already edited by a professional editing company.
Reviewer 2 Report
The authors have worked on addressing most of my concerns, in particular providing higher resolution images for all the IHC-related results; they have also quantitated most of their western blot-related results and provided additional information for the material and methods section.
However, there are still two major points that haven’t been addressed and this reviewer still considers these are important concerns that need to be taken care of. Therefore, these major points are listed again:
- Regarding the murine model characterization presented in figure 1, comparative H&E stain, as well as IHC of glycagon, CK-19, insulin, amylase, and β-catenin in human MCN tissue must be provided.
-The authors provided additional descriptions about β-catenin in human MCN tissue in the results section, supported by a reference by Makoto et al. Unfortunately, this additional information is not addressing my request, or it seems to be misplaced in the author’s response. The cited work from Makoto et al. could actually support my concern related to the possible role of –catenin in the stroma of these pancreata, but not validating the murine model with respect to the human disease as I requested. Therefore, the authors MUST provide H&E images of human MCN tissue and AT LEAST IHC images of LKB1, β-catenin, and ABC, to support the relevance of their murine model.
- The authors are urged to discuss the data obtained from the cytokines expression profiles presented, under the light of MCN progression to pancreatic cancer, and the possible impact of those on the altered Wnt signaling pathway observed in their model, and within the context of the human disease.
-The authors added some sentences to the corresponding results section which unfortunately are not satisfactory. They claimed the intention to investigate the potential roles of these cytokines in the pathogenesis of pancreatic MCNs in future work. Unfortunately, the additional in the text does not provide any substantial discussion regarding the cytokine array analysis, as requested. The authors remain merely describing their observations without giving a contextual interpretation to these results under the light of pro-tumor and anti-tumor cytokines driving the onset and progression of the disease. Since this analysis seems to be just a bag of immune molecules found randomly without any rationale to support it, the authors should remove these results unless they find a way to support the study of these cytokines and discuss IN DETAIL the implications of their findings.
Minor points that still need to be addressed:
- Regarding my point:
Support the in vitro results by evaluating if genetically silencing of LKB1 in a disease-relevant cell line, could recapitulate the observations described in the Pdx-1Cre LKB1L/Lmouse murine model when implanted in mice. Additionally, test if the double ablation of LKB1 + β-catenin prevents the formation of the previously mentioned pancreatic lesions.
The authors replied: We understand the reviewer's viewpoint here. In the present study, we later decided to use human PDAC cell lines for in vitro functional tests since primary cell culture didn’t work (very tough, almost impossible according to our experience) for the establishment of primary cell lines from Pdx-1Cre LKB1L/L cystic precancerous or benign tumor model.
-The authors should include this information in the corresponding results section.
- On page 12, line 421, it should read Collet and colleagues, instead of Louis and his colleagues.
Author Response
Reply: We thank the reviewer for the compliments and especially for the constructive suggestions. In the following, corrections were performed according to the suggestions by the reviewer.
Regarding the reviewer’s point:
Support the in vitro results by evaluating if genetically silencing of LKB1 in a disease-relevant cell line, could recapitulate the observations described in the Pdx-1Cre LKB1L/Lmouse murine model when implanted in mice. Additionally, test if the double ablation of LKB1 + β-catenin prevents the formation of the previously mentioned pancreatic lesions.
The authors replied: We understand the reviewer's viewpoint here. In the present study, we later decided to use human PDAC cell lines for in vitro functional tests since primary cell culture didn’t work (very tough, almost impossible according to our experience) for the establishment of primary cell lines from Pdx-1Cre LKB1L/L cystic precancerous or benign tumor model.
Reply: We thank the reviewer’s constructive suggestion. The double ablation of LKB1 + β-catenin to prevent the formation of pancreatic lesions will be done in future publication.
-The authors should include this information in the corresponding results section.
On page 12, line 421, it should read Collet and colleagues, instead of Louis and his colleagues.
Reply: This reviewer point is well taken.
Round 4
Reviewer 2 Report
I regret to mention that the authors have still not addressed one of the major points, mentioned several times in previous revisions:
- The authors must discuss the data obtained from the cytokines expression profiles under the light of MCN-to-pancreatic cancer progression, and the possible impact of those on the altered Wnt signaling pathway, in both the murine model and especially within the context of the human disease. Merely mentioning the reduction of CCL1, CCL2, 299 CCL27, CXCL16, LIX, TNF RI RII, and IL-4, after treatment of Pdx-1CreLKB1L/L mice with FH535, with no supporting rationale for assessing the presence of these inflammatory molecules, and lacking the discussion of these results findings, IT IS NOT ENOUGH for this reviewer.
There is also a minor point to address on page 9, line 301. Instead of chemokines and chemokines, it should read cytokines and chemokines.
Author Response
Dear Reviewer,
We thank the reviewer for his/her constructive and helpful comments, which have, in our view, significantly improved this manuscript. In this document, reviewers’ comments are shown below.
- The authors must discuss the data obtained from the cytokines expression profiles under the light of MCN-to-pancreatic cancer progression, and the possible impact of those on the altered Wnt signaling pathway, in both the murine model and especially within the context of the human disease. Merely mentioning the reduction of CCL1, CCL2, 299 CCL27, CXCL16, LIX, TNF RI RII, and IL-4, after treatment of Pdx-1CreLKB1L/L mice with FH535, with no supporting rationale for assessing the presence of these inflammatory molecules, and lacking the discussion of these results findings, IT IS NOT ENOUGH for this reviewer.
- Reply: we have added some sentences to discuss these results in the discussion section of this revised version.
There is also a minor point to address on page 9, line 301. Instead of chemokines and chemokines, it should read cytokines and chemokines.
Reply: Thank you. This point is well taken.